# Age and Sex-Related Differences in Teicoplanine Isoform Concentrations in SARS-CoV-2 Patients

**DOI:** 10.3390/life13091792

**Published:** 2023-08-22

**Authors:** Sarah Allegra, Francesco Chiara, Marina Zanatta, Giulio Mengozzi, Maria Paola Puccinelli, Silvia De Francia

**Affiliations:** 1Laboratory of Clinical Pharmacology “Franco Ghezzo”, Department of Clinical and Biological Sciences, University of Turin, S. Luigi Gonzaga Hospital, 10124 Orbassano, TO, Italy; 336124@edu.unito.it (F.C.); m.zanatta1@campus.unimib.it (M.Z.); silvia.defrancia@unito.it (S.D.F.); 2Laboratory of Clinical Biochemistry “Baldi e Riberi”, Metabolic Diseases Unit, AOU Città della Salute e della Scienza di Torino, 10126 Torino, TO, Italy; gmengozzi@cittadellasalute.to.it (G.M.); mpuccinelli@cittadellasalute.to.it (M.P.P.)

**Keywords:** COVID, sex, Targocid, gender, pediatrics

## Abstract

Teicoplanin, a glycopeptide antibiotic commonly used to treat bacterial infections, was discovered to be active in vitro against SARS-CoV-2. The aim of this study was to assess the levels of teicoplanin and its components in a cohort of adult and pediatric SARS-CoV-2 patients, evaluating the effect of sex and age on analyte concentrations. The levels of AST, ALT and leukocytes were shown to be higher in females, while the C reactive protein was higher in males. Evaluating the absence/presence of teicoplanin isoforms, we observed that A2-2_3 is the only one consistently present in pediatrics and adults. In adult men and all pediatrics, A2-4_5 is always present. In pediatrics, except for A3-1, median isoform concentrations were higher in females; on the contrary, in adult patients, males showed higher levels. This is the first study to describe levels of teicoplanin isoforms in SARS-CoV-2 infected patients in males and females, and pediatrics and adults, despite the small sample size of our cohort. The observed results imply that additional testing, via therapeutic drug monitoring, may be helpful to more effectively manage infections, particularly those caused by the most recent viruses.

## 1. Introduction

Millions of pneumonia cases have been reported worldwide since December 2019 due to a novel coronavirus, named severe acute respiratory syndrome coronavirus 2 (SARS-CoV-2), that has developed and spread internationally [1,2,3,4,5]. Laboratories and medical teams have focused their efforts on repurposing Food and Drug Administration (FDA)-approved medications to treat the most severe SARS-CoV-2 cases in the lack of a known verified effective therapy. Teicoplanin is a glycopeptide antibiotic commonly used to treat bacterial infections. Furthermore, it is a vancomycin-related glycopeptide antibiotic, which prevents Bacillus subtilis from synthesizing cell walls. This inhibition is accompanied by an intracellular buildup of UDP-N-acetyl-muramyl-pentapeptide. Teicoplanin inhibits peptidoglycan synthesis in a cell-free Bacillus stearothermophilus system by 50% at 40 micrograms/mL and 100% at 100 micrograms/mL; the lipid intermediate accumulates concurrently as the peptidoglycan synthesis is suppressed. Teicoplanin interacts with N,N’-diacetyl-L-lysyl-D-alanine to generate a complex when it attaches to cell walls. By using spectrophotometric titration, the association constant of this compound was determined to be 2.56 × 10^6^ L·mol^−^^1^ [6,7,8,9]. Teicoplanin was discovered to be active in vitro against SARS-CoV and has since been included in the list of compounds that could be utilized in the therapeutic toolkit against COVID-19 [10]. This antibiotic, which is currently used to treat Gram-positive bacterial infections, particularly staphylococcal infections, has already demonstrated effectiveness against a number of viruses, including the Ebola virus, influenza virus, flavivirus, hepatitis C virus, human immunodeficiency virus (HIV), and the Middle East respiratory syndrome coronavirus (MERS-CoV), among others [11,12].

Teicoplanin inhibits the low-pH cleavage of the viral spike protein by cathepsin L in the late endosomes, which stops the release of genomic viral RNA and the continuation of the virus replication cycle, according to Zhou et al. [12]. Teicoplanin acts on an early stage of the viral life cycle in coronaviruses. The target region that acts as the cleavage site for cathepsin L is conserved among the SARS-CoV-2 spike protein, according to a recent work by the same scientists, which demonstrated that this activity was conserved against SARS-CoV-2 [13]. Teicoplanin in vitro inhibitory concentration (IC 50) was 1.66 M, which is a lot lower than the concentration that may be found in human blood at 400 mg dose per day (8.78 M) [11].

Teicoplanin consists of a mixture of six major components (A2-1, A2-2, A2-3, A2-4, A2-5 and A3-1), one of which has an oxygen-bonded hydrogen terminal and five of which have an R-substituent and decanoic acid; and four minor components (RS-1, R2-2, RS-3 and RS-4). The components are found in different proportions and vary depending on the growth medium of the producing bacterium; each requires different environments for growth, e.g., component A2-1 requires the presence of linoleic acid, while A2-2 needs oleic acid and as for A2-4 and A2-5, the presence of valine, leucine or isoleucine [14].

Due to its pharmacokinetic properties similar to vancomycin, teicoplanin is used as an antimicrobial agent in adults, children and infants.

The purpose of this study is to assess the levels of teicoplanin and its components in a cohort of adult and pediatric SARS-CoV-2 patients, evaluating the effect of sex and age on analyte concentrations.

## 2. Materials and Methods

### 2.1. Patients and Inclusion Criteria

We performed a retrospective study in a cohort of SARS-CoV-2 patients treated at the Regina Margherita Hospital and Molinette Hospital in Turin. Teicoplanin levels were determined between December 2020 and March 2022.

Teicoplanin (Targocid^®^ 400 mg; Sanofi, Diegem, Belgium) is recommended to be administered at a dose of 10–15 mg/kg over 60 min, three times within a 12 h period to begin with, and then once every 24 h after. Teicoplanin was administered intravenously over 3–30 min using a calibrated syringe.

Driver patients who had at least one treatment episode with teicoplanin with at least one plasma determination were included in the study. Collected data were age (years) and sex (male and female), alanine aminotransferase (ALT; mg/dL), aspartate aminotransferase (AST; mg/dL), leucocytes count (number * 10^9^/L), c reactive protein (CRP; mg/dL).

The study protocol (“Studio retrospettivo per la valutazione farmacocinetica e farmaco-genetica della terapia antibiotica con glicopeptidi”) was approved by the local Ethics Committee. A written informed consent for the study was obtained from each enrolled subject.

### 2.2. HPLC Analysis

Plasma A3-1, A2-1, A2-2_3 and A2-4_5 analyte concentrations were determined from plasma samples obtained at the end of dosing interval, before the next drug-dose intake.

Patient blood samples were collected in the lithium-heparin tube and centrifuged at 1500 rpm for 10 min at 4 °C. Analyte quantification was performed by High-Performance Liquid Chromatography (HPLC), equipped with a binary pump system (Shimadzu Corporation, Kyoto, Japan) coupled to a triple quadrupole mass spectrometer (SCIEX-triple quad 4500 MD; AB SCIEX instruments, Old Connecticut Path Framingham, MA, USA). The chromatographic separation was realized using a Mediterranea sea18 (Teknokroma, Barcelona, Spain) column. Two mobile phases were used for the chromatographic run:

Phase A: 5 mM ammonium formate water +0.01% *v*/*v* formic acid.

Phase B: Methanol/Isorpopanol 8:1 in 5 mM ammonium formate water + 0.01% *v*/*v* formic acid.

The analysis was carried out at the constant flow rate of 0.3 mL/min. The chromatographic gradient is resumed in Table 1.

The data obtained were processed with Analyst^®^ software (Sciex, Milan, Italy, https://sciex.com/products/software/analyst-software, accessed date 10 August 2023) and quantification with Multiquant software (Sciex, Milan, Italy, https://sciex.com/products/software/multiquant-software, accessed date 10 August 2023). Internal standard quantification was used, fitted with linear regression.

### 2.3. Statistical Analysis

For descriptive statistics, continuous and non-normal variables were summarized as median values (considering all the enrolled patients) and the interquartile range (QRange, difference between upper and lower quartiles: Q3–Q1) was calculated to measure the statistical dispersion of the data; categorical variables were described as frequency and percentage. The Pearson linear correlation coefficient (r) was performed to investigate the strength of the association between continuous variables (level of statistical significance *p*-value < 0.05). The following subgroups were considered: male, female, adults and pediatrics. Any predictive power of the considered variables on teicoplanin isoform levels was finally evaluated through univariate and multivariate linear regression analysis (level of statistical significance *p*-value < 0.05). All tests were performed with SAS version 9.4 (SAS Institute, Cary, NC, USA).

## 3. Results

### 3.1. Study Population

We retrieved the data of 41 patients, 25 males and 16 females. Baseline characteristics of enrolled patients, considering male and female patients of pediatric and adult cohorts, are listed in Table 2 and represented in Figure 1. No significative differences were observed between ALT, AST, leukocytes count and CRP levels in the evaluated groups.

### 3.2. Effect of Age Ang Gender on Teicoplanin Isoforms

As shown in Figure 2, panel A, isoform A2-1 is absent in 4% (1/25) of male patients and 12.5% (2/16) of female patients; isoform A2-2_3 is quantifiable in all male and female patients; isoform A2-4_5 is absent in 6.25% (1/16) of female patients; isoform A3-1 is absent in 24% (6/26) of male patients, and in 37.5% of females (6/16).

Considering adult patients (Figure 2, panel B), isoform A2-1 is absent in 5% (1/20) of male patients, and 16.6% (2/12) in females; isoform A2-2_3 is quantifiable in all male and female patients; isoform A2-4_5 is absent in 8.3% (1/12) of female patients; isoform A3-1 is absent in 20% (4/20) of male patients, and in 41.6% (5/12) of females.

Evaluating only pediatrics (Figure 2, panel C), we observed that isoforms A2-1, A2-2_3 and A2-4_5 are quantifiable in all male and female patients; isoform A3-1 is absent in 40% (2/5) of male patients, and in 25% (1/4) of female patients.

Considering the presence of each drug isoform, the pharmacokinetic characteristics of enrolled patients, splitting male and female patients of pediatric and adult cohorts, are listed in Table 3 and represented in Figure 3.

No significative correlation was observed between the A3-1, A2-1, A2-2_3 and A2-4_5 concentrations in the evaluated groups. The regression model showed no predictive role of age and sex on teicoplanin isoform levels.

### 3.3. Correlation between ALT, AST, Leukocytes Count and CRP and Teicoplanin Isoforms

No significative correlation has was observed between the A3-1, A2-1, A2-2_3 and A2-4_5 concentrations and ALT, AST, leukocyte count and CRP. Regression model showed no predictive role of ALT, AST, leukocyte count and CRP on teicoplanin isoform levels.

## 4. Discussion

Recent studies have revealed that bacterial infections are the primary cause of death in patients admitted to intensive care units and in the post-operative period if they are not treated properly. Bacterial infections remain an ongoing scientific interest, particularly in the hospital context. However, inappropriate use of antibiotics is the primary contributor to a contemporary issue for which the scientific community is urged to find a prompt remedy, namely the antibiotic resistance. Antibiotic resistance is one of the biggest threats to global health. It can affect anyone, of any age, in any country, occurring naturally, but misuse of antibiotics in humans and animals is accelerating the process. Antibiotic resistance leads to longer hospital stays, higher medical costs and increased mortality [15]. Then, due to the growing prevalence of resistance to conventional antibiotics, glycopeptides are now more often employed alone or in combination with other substances to treat bacterial infections brought on by Gram-positive bacteria. Teicoplanin, which has been the focus of numerous studies and whose therapeutic monitoring is crucial, is one of the most commonly used glycopeptide medications.

The aim of the study was to evaluate teicoplanin levels in a cohort of SARS-CoV-2 pediatric and adult patients with special respect to sex-and age-related differences. Gonadal hormones largely affect the form and function of gender-specific organs and play a role in the development of gender-specific features. There are differences between men and women in a wide range of traits due to their role in controlling the construction and function of almost every tissue and organ in the mammalian body, according to studies. Sex hormones affect the development of both male and female-specific features and are linked to several processes of reproduction, differentiation, development, growth, and homeostasis. Men and women differ in a number of organs, including brain, bones, liver, and kidneys [16]. Estrogen has a well-established protective role in health and disease, and it is the major cause of gender differences in a number of biochemical indicators. There are a few studies that compare liver parameters in both sexes available [17,18].

The levels of AST and ALT were shown to be higher in females, both in adults and pediatrics. Contrary to what Mera et al. [18] reported, who suggested that women had reduced AST and ALT levels, we observed higher liver transferases in female patients. In different studies, females were found to have high-ALT levels [19,20,21]. The hormonal state and differences in muscle mass are thought to be the causes of this variance in liver enzymes [20]. However, it may be difficult to determine how female gonadotropins affect the biochemistry of the liver. The impact of estradiol supplementation on liver indicators was investigated by Moore et al. [22]. They discovered no changes in the levels of bilirubin or liver enzymes after hormone replacement therapy. With and without estrogen-containing hormone replacement therapy, postmenopausal women’s liver profiles were evaluated by Crippin and colleagues [23]. They stated that women receiving hormone replacement therapy did not have elevated bilirubin levels. Guattery and colleagues provided contradictory data, stating elevated bilirubin levels in biliary cirrhotic women receiving hormone therapy that required medication discontinuation [24]. Considering the setting of our study, despite the fact that SARS-CoV-2 mostly damages the lungs and respiratory system [2,25,26] growing evidence suggests that the liver may also be affected, leading to liver damage [27,28,29]. The potential pathophysiology of liver damage in people infected with SARS-CoV-2 is complicated. It might be connected to liver damage brought on by drugs, hypoxia-ischemic damage, hyper-inflammatory cytokine storm, and direct viral injury [30]. The 14.8 to 53.0% of SARS-CoV-2 infected patients have reported having liver damage [31,32,33,34]. Considering leukocytes, we reported higher levels in females. Immune cells are activated once the human body becomes infected, and cytokines follow in an effort to fight the virus [35]. It is becoming more widely acknowledged that COVID-19 affects men more severely than women [36,37]. Male patients are more likely to be hospitalized, have a more severe course of disease, and have a higher fatality rate, according to preliminary investigations, even if infection frequency is the same in men and women [38,39]. Sex differences in innate and adaptive immunity are well documented, and the resulting sex-specific responses to vaccinations and infections have been proposed as a major factor in the varied reactions between sexes to COVID-19 [40]. Although the innate immune response appeared to be more active in male patients, as seen by greater levels of circulating inflammatory neutrophils and monocytes, the adaptive immune response was more potent in females, and females had considerably more B cells at 7 days and 14 days [41], a fact maybe correlated with Long-COVID, more frequent in female patients. On the contrary, we showed lower CRP in females. While some claim elevated CRP levels increase the risk of death in male patients [42], other research suggests that biomarkers of inflammation may be more accurate in predicting death in female patients [43].

Evaluating the absence/presence of teicoplanin isoforms, we observed that A2-2_3 is the only one always present in pediatrics and adults; in pediatrics, A3-1 is the only one absent in a percentage of male and female patients. In adult men and in all pediatrics, A2-4_5 is always present. In pediatrics, except for A3-1, median isoform concentrations were higher in females; on the contrary, in adult patients, males showed consistently higher drug levels. A higher clearance in younger children leads to lower levels than in adults [44,45,46].

Lukas et al. conducted a population pharmacokinetics analysis on 20 infants and children aged 4 months to 10 years. They revealed that 8% of young children aged < 12 months had TTLs < 10 μg/mL compared to 35% of children aged ≥12 months [40]. In another analysis, Ramos–Martin and colleagues created a pharmacokinetics model of teicoplanin based on data from 39 children and 33 adults. They reported that the rate of predicted subtherapeutic concentrations of <10 μg/mL increased with increasing weight/age from 39% (in children with a presumed body weight of 10 kg) to 55% (in children with a presumed body weight of 25 kg) to 70% (in children with a presumed body weight of 50 kg) and 75% (in adults) [47]. Other researchers either failed to find any statistically significant correlation between teicoplanin pharmacokinetics and age (13 children between the ages of 2 and 12), or they only found a non-significant trend (12 children between the ages of 2.4 and 11) [45]. Also, Tarral et al. observed that clearance was higher in newborns when they compared six children with a mean age of 7 years with four neonates with a mean age of 8.5 days [48]. On the sex-related variations in teicoplanin pharmacokinetics, there is relatively little information available in the current research. According to Matthews et al., who examined teicoplanin levels in 141 persons with bone and joint infections, men had lower drug levels than women [49]. Teicoplanin is distributed differently in various tissues, with skeletal muscles having the highest quantities and adipose tissue having the lowest concentrations.

## 5. Conclusions

This is the first study created to individually investigate the levels of teicoplanin isoforms in SARS-CoV-2 infected patients in males and females, and pediatrics and adults, despite the small sample size of our cohort. The observed results imply that additional testing of this drug’s therapeutic monitoring may be helpful to more effectively manage infections, particularly those caused by the most recent viruses. The study limitation must be highlighted: first, the small sample size, and then the lack of hormonal and viral information of the enrolled patients. To better understand the pharmacokinetics of teicoplanin, more research is required with a bigger cohort and data on the hormonal phases, body weight, and body fat distribution of both men and women.

Drug repurposing should be more tailored in order to be sufficiently effective.

## Figures and Tables

**Figure 1 life-13-01792-f001:**
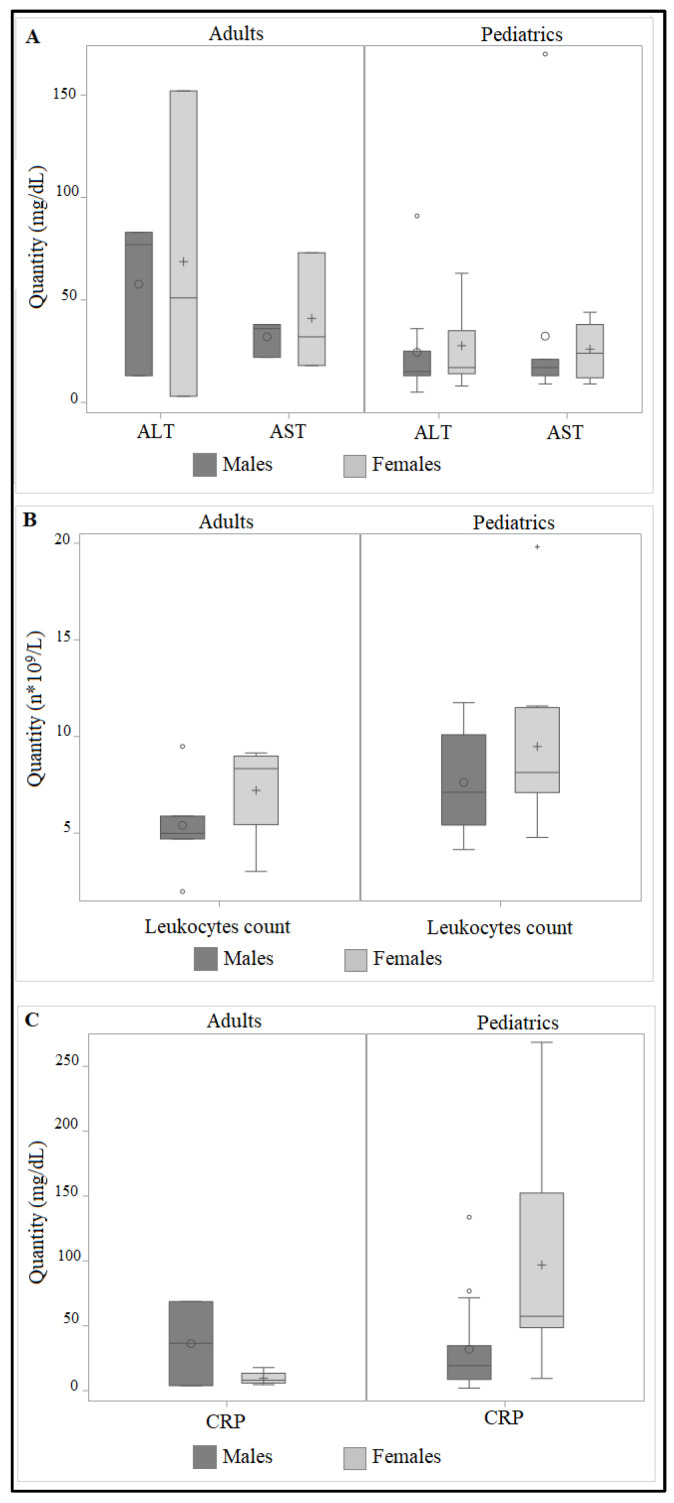
Distribution of alanine aminotransferase (ALT; panel (**A**)), aspartate aminotransferase (AST; panel (**A**)), leukocytes count (panel (**B**)) and c reactive protein levels (CRP; panel (**C**)) in male (dark grey) and female (light grey) patients, considering pediatric and adult cohorts. Boxes and black lines in boxes represent, respectively, interquartile ranges (IQR) and median values; open dots represent mean values. Median values (horizontal line), IQR (bars), highest and lowest value (whiskers) are shown.

**Figure 2 life-13-01792-f002:**
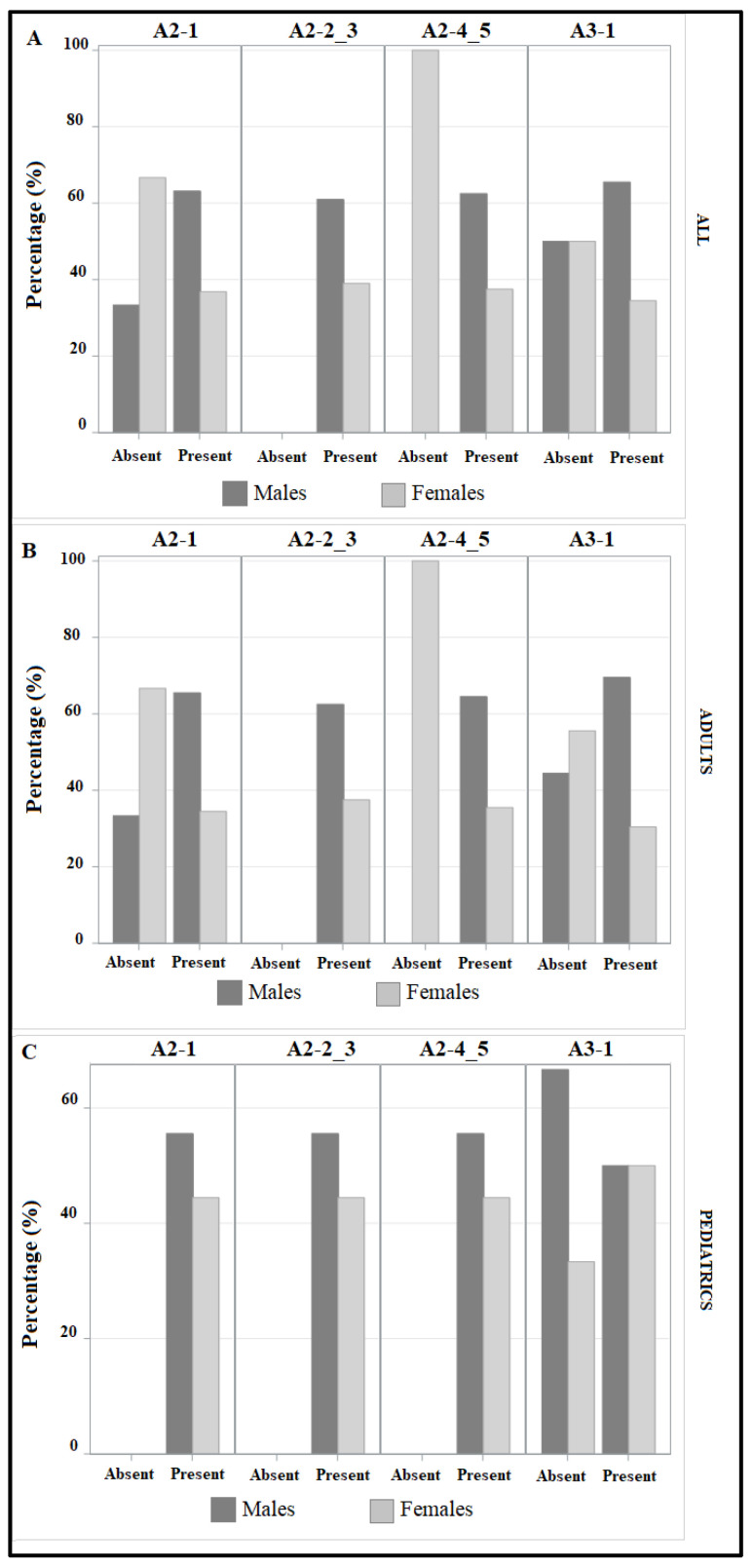
Frequencies of teicoplanin isoforms A3-1, A2-1, A2-2_3 and A2-4_5 in male (dark grey) and female (light grey) patients, considering all the enrolled patients (panel (**A**)), adults (panel (**B**)) and pediatrics (panel (**C**)).

**Figure 3 life-13-01792-f003:**
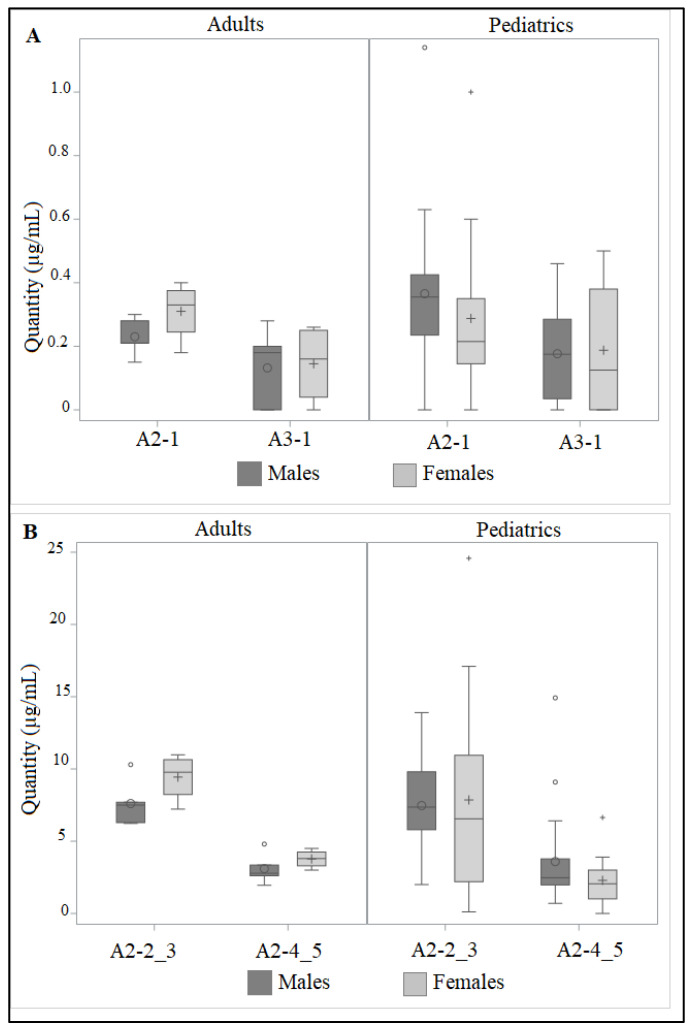
Distribution of teicoplanin isoforms A3-1, A2-1, (panel (**A**)) A2-2_3 and A2-4_5 (panel (**B**)) in male (dark grey) and female (light grey) patients, considering pediatric and adult cohorts. Boxes and black lines in boxes represent, respectively, interquartile ranges (IQR) and median values; open dots represent mean values. Median values (horizontal line), IQR (bars), highest and lowest value (whiskers) are shown.

**Table 1 life-13-01792-t001:** Chromatographic gradient.

Time	Flow (mL/min)	Mobile Phase A	Mobile Phase B
0.01	0.3	95	5
3.00	0.3	95	5
4.50	0.3	38	62
5.50	0.3	38	62
7.50	0.3	25	75
8.00	0.3	25	75
8.50	0.3	0	100
9.00	0.3	0	100
9.50	0.3	95	5

**Table 2 life-13-01792-t002:** Demographic and clinical characteristics of the enrolled patients, considering female and male patients in pediatric and adult cohorts. List of abbreviations: QRange, difference between upper and lower quartiles: Q3-Q1; ALT, alanine aminotransferase; AST, aspartate aminotransferase; CRP, c reactive protein.

		Pediatrics		Adults	
		F	M	F	M
Age (years)	Median	13.50	14.00	81.00	63.00
	QRange	10.50	4.00	9.00	20.00
ALT (mg/dL)	Median	51.00	77.00	17.00	15.00
	QRange	149.00	70.00	21.00	12.00
AST (mg/dL)	Median	32.00	36.00	24.00	17.00
	QRange	55.00	16.00	26.00	8.00
LEUKOCYTES COUNT (number * 10^9^/L)	Median	8.34	4.99	8.14	7.12
	QRange	3.55	1.18	4.40	4.67
CRP (mg/dL)	Median	8.05	36.50	57.30	19.30
	QRange	7.55	64.90	103.70	26.00

**Table 3 life-13-01792-t003:** Pharmacokinetics characteristics of the enrolled patients, considering female and male patients in pediatric and adult cohorts. List of abbreviations: QRange, difference between upper and lower quartiles.

ISOFORM (µg/mL)		Pediatrics		Adults	
		F	M	F	M
A2-1	Median	0.33	0.21	0.22	0.36
	QRange	0.13	0.07	0.21	0.19
A2-2_3	Median	9.78	7.50	6.55	7.36
	QRange	2.41	1.41	8.75	4.00
A2-4_5	Median	3.80	2.78	2.06	2.48
	QRange	0.95	0.74	1.99	1.80
A3-1	Median	0.16	0.18	0.13	0.18
	QRange	0.21	0.20	0.38	0.2

## Data Availability

Not applicable.

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
