# Peer review of "Age and Sex-Related Differences in Teicoplanine Isoform Concentrations in SARS-CoV-2 Patients"

_life, 2023, doi:10.3390/life13091792_

Round 1
Reviewer 1 Report
I do not have any concerns about the manuscript. In fact, repurposing known antibacterials for antiviral application is quite a “hot” topic in post-covid era. The work has nice language, methodology is sound. I believe the manuscript would be a nice contribution to the journal. A single minor comment is that authors might also cite some general reviews about the biology of teicoplanin in case readers would like to have any other ideas about antibiotic except medicinal aspects.
Author Response
Reviewer 1
Comments and Suggestions for Authors
I do not have any concerns about the manuscript. In fact, repurposing known antibacterials for antiviral application is quite a “hot” topic in post-covid era. The work has nice language, methodology is sound. I believe the manuscript would be a nice contribution to the journal. A single minor comment is that authors might also cite some general reviews about the biology of teicoplanin in case readers would like to have any other ideas about antibiotic except medicinal aspects.
Dear Reviewer, thank you very much for the thorough review. As you suggest we added the following reviews to our paper:
It is vancomycin-related glycopeptide antibiotic, prevents Bacillus subtilis from synthe-sizing cell walls. This inhibition is accompanied by an intracellular buildup of UDP-N-acetyl-muramyl-pentapeptide. Teicoplanin inhibits peptidoglycan synthesis in a cell-free Bacillus stearothermophilus system by 50% at 40 micrograms/ml and 100% at 100 micrograms/ml; the lipid intermediate accumulates concurrently as the peptidogly-can synthesis is suppressed. Teicoplanin interacts with N,N'-diacetyl-L-lysyl-D-alanine to generate a complex when it attaches to cell walls. By using spectrophotometric titration, the association constant of this compound was determined to be 2.56 X 10(6) liters mol-1 [6-9].
- Yushchuk, O.; Ostash, B.; Truman, A.W.; Marinelli, F.; Fedorenko, V. Teicoplanin biosynthesis: unraveling the interplay of structural, regulatory, and resistance genes. Appl Microbiol Biotechnol 2020, 104, 3279-3291, doi:10.1007/s00253-020-10436-y.
- Boger, D.L. Vancomycin, teicoplanin, and ramoplanin: synthetic and mechanistic studies. Med Res Rev 2001, 21, 356-381, doi:10.1002/med.1014.
- Butler, M.S.; Hansford, K.A.; Blaskovich, M.A.; Halai, R.; Cooper, M.A. Glycopeptide antibiotics: back to the future. J Antibiot (Tokyo) 2014, 67, 631-644, doi:10.1038/ja.2014.111.
- Coronelli, C.; Gallo, G.G.; Cavalleri, B. Teicoplanin: chemical, physico-chemical and biological aspects. Farmaco Sci 1987, 42, 767-786.

Reviewer 2 Report
Overall, I am not recommended for this study to be published in the highest quartile journal because the results presented in the manuscript are only superficial research findings and the data presented are insufficient. An extensive study needs to be done first before it is eligible to be published in the high ranked journals.
Extensive editing of English language required.
Author Response
Reviewer 2
Overall, I am not recommended for this study to be published in the highest quartile journal because the results presented in the manuscript are only superficial research findings and the data presented are insufficient. An extensive study needs to be done first before it is eligible to be published in the high ranked journals.
Extensive editing of English language required.
Dear Reviewer,
Our manuscript has been checked by a native English-speaking colleague.
Thank you very much for the thorough review. We are very sorry to read your opinion about our work. The article has been revised, also based on the suggestions of the other reviewers, who have had a positive opinion on our work. We are proposing a pilot study on a topic of great interest, highlighting those that we have observed in our population, albeit not numerous, of COVID positive patients treated with teicoplanin. It is certainly of collateral impact compared to other studies on COVID, for example compared to those concerning the development of vaccines. However, ours remains an important observation on prosed infection therapy. We accept your negative comment, but we would like to reiterate that, according to us, it remains important to enrich the literature also with this information on COVID treatment, with particular attention to differences in age and biological sex.

Reviewer 3 Report
One of the methods of treating diseases is to try to use already registered drugs. Teicoplanin, like the most popular of this group of drugs, vancomycin, belongs to glycopeptide antibiotics. Discovered in 1978, ten years later it came into use in Europe. It is mainly used to fight infections with Gram-positive bacteria. Recently, there have been reports of its activity against some viruses. The reviewed article describes studies on the level of leukocytes and aminotransferases in patients who were administered teicoplanin.
The introduction contains the most important information for the subject of the article and was written based on the latest literature. The research was designed and conducted correctly, and the conclusions are beyond doubt.
I agree with the authors that teicoplanin may prove to be an interesting therapeutic not only in bacterial infections but also in those of viral origin. However, this requires more extensive research on a larger group of patients, which the presented research results may be a good start.
Before publication, authors should supplement the description of HPLC analyzes by adding: column parameters, eluent composition and flow rate. Such information is extremely important for the possibility of repeating the analyzes by others.
Author Response
Reviewer 3
Comments and Suggestions for Authors
One of the methods of treating diseases is to try to use already registered drugs. Teicoplanin, like the most popular of this group of drugs, vancomycin, belongs to glycopeptide antibiotics. Discovered in 1978, ten years later it came into use in Europe. It is mainly used to fight infections with Gram-positive bacteria. Recently, there have been reports of its activity against some viruses. The reviewed article describes studies on the level of leukocytes and aminotransferases in patients who were administered teicoplanin.
The introduction contains the most important information for the subject of the article and was written based on the latest literature. The research was designed and conducted correctly, and the conclusions are beyond doubt.
I agree with the authors that teicoplanin may prove to be an interesting therapeutic not only in bacterial infections but also in those of viral origin. However, this requires more extensive research on a larger group of patients, which the presented research results may be a good start.
Before publication, authors should supplement the description of HPLC analyzes by adding: column parameters, eluent composition and flow rate. Such information is extremely important for the possibility of repeating the analyzes by others.
Dear Reviewer,
thank you very much for the thorough review. We agree to all specific comments addressed and have revised our paper in light of the useful suggestions. Answers to the specific comments/suggestions/queries are as follows.
- The study limitation section has been improved, with your suggestions.
- The chromatographic separation has been realized using a Mediterranea sea18 (Teknokroma, Barcelona, Spain) column. For the chromatographic run has been used two mobile phases:
Phase A: 5mM ammonium formate water + 0.01% v/v formic acid
Phase B: Methanol/Isorpopanol 8:1 5mM ammonium formate + 0.01% v/v formic acid.
The analysis was carried out at the constant flow rate of 0.3 mL/min. The chromatographic gradient is resumed in Table 1.
Table 1. Chromatographic gradient.
|
Time |
Flow (mL/min) |
Mobile Phase A |
Mobile Phase B |
|
||
|
0.01 |
0.3 |
95 |
5 |
|||
|
3.00 |
0.3 |
95 |
5 |
|||
|
4.50 |
0.3 |
38 |
62 |
|||
|
5.50 |
0.3 |
38 |
62 |
|||
|
7.50 |
0.3 |
25 |
75 |
|||
|
8.00 |
0.3 |
25 |
75 |
|||
|
8.50 |
0.3 |
0 |
100 |
|||
|
9.00 |
0.3 |
0 |
100 |
|||
|
9.50 |
0.3 |
95 |
5 |
|||

Reviewer 4 Report
In this manuscript the authors performed a retrospective study in a cohort of SARS-CoV-2 patients, and explored whether there's a correlation between the teicoplanine isoform concentrations and the age/sex of the patients. Given the lack of data for drug monitoring of teicoplanine, I believe this report will be valuable for optimizing therapy in certain patients. I've also made some in-line comments and edits in the attached file for the authors to address.

English language is generally fine, but some minor edits are required.
Author Response
Reviewer 4
Comments and Suggestions for Authors
In this manuscript the authors performed a retrospective study in a cohort of SARS-CoV-2 patients, and explored whether there's a correlation between the teicoplanine isoform concentrations and the age/sex of the patients. Given the lack of data for drug monitoring of teicoplanine, I believe this report will be valuable for optimizing therapy in certain patients. I've also made some in-line comments and edits in the attached file for the authors to address.
Dear Reviewer,
thank you very much for the thorough review. We agree to all specific comments addressed and have revised our paper in light of the useful suggestions. Answers to the specific comments/suggestions/queries are as follows:
- Thank you for your professional comment, results have been modified as you suggested.
- The entire text has been revised as you suggested, the changes in the text have been marked in yellow.

Round 2
Reviewer 2 Report
Iam not recommend this study. Not achieved the scientific level writing. Major improvised are needed.
-